# Searching to Exploit Memorization Effect in Learning from noisy Labels

## Abstract

Sample-selection approaches, which attempt to pick up clean instances from the noisy training data set, have become one promising direction to robust learning from noisy labels. These methods all build on the memorization effect, which means deep networks learn easy patterns first and then gradually over-fit the training data set. In this paper, we show how to properly select instances so that the training process can benefit the most from the memorization effect is a hard problem. Specifically, memorization can heavily depend on many factors, e.g., data set and network architecture. Nonetheless, there still exists general patterns of how memorization can occur. These facts motivate us to exploit memorization by automated machine learning (AutoML) techniques. First, we design an expressive but compact search space based on observed general patterns. Then, we propose to use the natural gradient-based search algorithm to efficiently search through space. Finally, extensive experiments on both synthetic data sets and benchmark data sets demonstrate that the proposed method can not only be much efficient than existing AutoML algorithms but can also achieve much better performance than the state-of-the-art approaches for learning from noisy labels.

## 1 Introduction

Learning with deep neural networks has enjoyed huge empirical success in recent years across a wide variety of tasks, from image processing to speech recognition, and from language modeling to recommender system (Goodfellow et al., 2016). However, their success highly counts on the availability of well-annotated and big data, which is barely available for real-world applications. Instead, what we are facing with in practice are large data sets which are collected from crowd-sourcing platforms or crawled from the Internet, thus containing many noisy labels (Li et al., 2017b; Patrini et al., 2017). Besides, due to the vast learning capacity of deep networks, they will eventually over-fit on these noisy labels, leading to poor predicting performance, which can be worse than that obtained from simple models (Zhang et al., 2016; Arpit et al., 2017).

To reduce negative effects from noisy labels, many methods have been proposed (Sukhbaatar et al., 2015; Reed et al., 2015; Patrini et al., 2017; Ghosh et al., 2017; Malach & Shalev-Shwartz, 2017) . Recently, a promising direction is training networks only on *selected instances* that are more likely to be clean (Jiang et al., 2018; Han et al., 2018b; Ma et al., 2018; Yu et al., 2019; Wang et al., 2019). Intuitively, as the training data becomes less noisy, better performance can be obtained. Among those works, the representative methods are MentorNet (Jiang et al., 2018) and Co-teaching (Han et al., 2018b; Yu et al., 2019), they take small-loss samples in each mini-batch as clean instances. Specifically, MentorNet pre-trains an extra network, and then uses the extra network for selecting clean instances to guide the training. When the clean validation data is not available, MentorNet has to use a predefined curriculum (Bengio et al., 2009). Co-teaching is an improvement over MentorNet, it simultaneously maintains two networks which have identical architectures during the training process. And in each mini-batch of data, each network is updated using *the other* network's small-loss instances.

To the success of these sample-selection methods, the memorization effect of deep networks (Zhang et al., 2016; Arpit et al., 2017) is the crux. Memorization happens widely in various architectures of deep network, e.g., multilayer perceptron (MLP) and convolutional neural network (CNN). Specifically, it means that deep networks tend to learn easy and correct patterns first and then over-fit on

(possibly noisy) training data set (see Fig.1(a)-(b)). Thus, when learning with noisy labels, while the validation loss will first increase and then significantly decrease, the training loss will continuously get smaller with more training epochs.

Due to such effect, sample-selection methods can learn correct patterns at early stage and then use the obtained discriminative ability to filter out corrupted instances in subsequent training epochs (Jiang et al., 2018; Han et al., 2018b; Chen et al., 2019). While the memorization effect is critical to the success of sample-selection methods, however, how to properly exploit it is not addressed in the literature. And trivial attempts can easily lead to even worse performance than standard deep networks (Han et al., 2018b). Some recent endeavors seek to evade from this problems by integrating with other auxiliary information, e.g., a small clean subset is used in (Ren et al., 2018), and knowledge graphs are utilized in (Li et al., 2017b).

In this paper, motivated by the success of automated machine learning (AutoML) on designing data-dependent models (Hutter et al., 2018), and the fact that memorization heavily depends on many factors (Zhang et al., 2016; Arpit et al., 2017), we propose to exploit memorization effects automatically using AutoML techniques. Contributions are summarized as follows:

- First, to have an in-depth understanding of why it is difficult to tune sample-selection methods with good performance. We examine behaviors of memorization effect from multiple perspectives. We find that, while there exist general patterns in how memorization occurs with the training process (see Fig.1(a)-(b)), it is hard to quantize to which extend such effect can happen (see Fig.1(b)-(f)). Especially, memorization can be affected by many factors, e.g., data sets, network architectures, and the choice of the optimizers. It is exactly such complex dependency make the design of proper sample-selection rules a hard problem, which motivates us to solve the problem by AutoML techniques.

- To make good use of AutoML techniques, we then derive an expressive search space for exploiting memorization, which is from the above observations, i.e., the curvature of how many instances need to be sampled during iterating should be similar with the inverse of the learning curve on the validation set. Such a space is not too huge since it has only a few variables, thus allows subsequent algorithms converging fast to promising candidates.

- Then, to design an efficient algorithm, we show the failure of gradient-based methods and the inefficiency of derivative-free methods. These motivate us to take a probabilistic view of the search problem and adopt natural gradient descent (Amari, 1998; Pascanu & Bengio, 2013) for optimization. The designed algorithm can effectively address above problems and is significantly faster than other popular search algorithms.

- Finally, we conduct extensive experiments on both synthetic, benchmark, and real data sets, under various settings using different network architectures. These experiments demonstrate that the proposed method can not only be much more efficient than existing AutoML algorithms, but also can achieve much better performance than the state-of-the-art sample-selection approaches designed by humans. Besides, we further visualize and explain the searched functions, which can also help design better rules to control memorization effects in the future.

## 2 RELATED WORK

### 2.1 LEARNING FROM NOISY LABELS

The mainstream research focuses on *class-conditional noise* (CCN) (Angluin & Laird, 1988), where the label corruption is independent of features. Generally, recent methods for handling CCN model can be classified into three categories. The first one is based on the estimation of transition matrix, which tries to capture how correct labels flip into wrong ones (Sukhbaatar et al., 2015; Reed et al., 2015; Patrini et al., 2017; Ghosh et al., 2017). These methods then use the estimated matrix to correct gradients or loss during training. However, they are fragile to heavy noise and unable to handle many classes (Han et al., 2018b). The second type is the regularization approach (Miyato et al., 2016; Laine & Aila, 2017; Tarvainen & Valpola, 2017). Although regularization approach can achieve a satisfying performance, it is still an *incomplete* approach since (Jiang et al., 2018) shows that it can only *delay* the overfitting progress rather than *avoid* it, i.e. given enough training time, it can still fit the noisy data completely. Thus, it requires much domain knowledge to determine the appropriate

number of training epochs in order to prevent overfitting. The last one is sample-selection approach, which attempts to reduce negative effects from noisy labels by selecting clean instances during training. The recent state-of-the-art method is also built on sample-selection approach (Jiang et al., 2018; Han et al., 2018b; Malach & Shalev-Shwartz, 2017; Yu et al., 2019).

Active learning (Settles, 1994) is a closely related method, which iteratively selects unlabeled samples with high high-confident predictions into the training data set. Thus, to do active learning, we need to obtain a classifier of which the performance is good enough. As a result, active learning is not applicable for directly learning from noisy labels here.

A promising criteria to select "clean instances" is to pick up instances that has relatively small losses in each mini-batch (Jiang et al., 2018; Han et al., 2018b). The fundamental property behind these methods is the memorization effect of deep networks (Zhang et al., 2016; Arpit et al., 2017), which means deep networks can learn simple patterns first and then start to over-fit. Such effect helps classifiers set up discriminate ability in the early stage, then make clean instances more likely to have smaller loss that those noisy ones. The general framework of sample-selection approach is in Alg.1. Specifically, some small-loss instances $\bar{\mathcal{D}}_f$ are selected from the mini-batch $\bar{\mathcal{D}}$ in step 5. These "clean" instances are then used to update network parameters in step 6. The $R(t)$ in step 8, which controls how many instances to be kept in each epoch, is the most important hyper-parameter as it explicitly exploits the memorization effect (Han et al., 2018b; Jiang et al., 2018; Yu et al., 2019).

---

**Algorithm 1** Framework of the sample-selection approach (Jiang et al., 2018; Han et al., 2018b).

1: **for** $t = 1, \cdots, T$ **do**
2:     shuffle training set $\mathcal{D}$;
3:     **for** $n = 1, \cdots, N$ **do**
4:         draw a mini-batch $\bar{\mathcal{D}}$ from $\mathcal{D}$;
5:         select $\bar{\mathcal{D}}_f$, i.e., $R(t)$ small-loss instances from $\bar{\mathcal{D}}$ based on network's predictions;
6:         update the network's parameter using gradient from $\bar{\mathcal{D}}_f$;
7:     **end for**
8:     exploit memorization effects using $R(t)$ *(estimation on percentage of clean instances)*;
9: **end for**

---

However, it is hard to exactly determine *how much* proportion of small-loss samples should be selected in each epoch (Jiang et al., 2018; Ren et al., 2018). As will be discussed in Sec.3.1, due to various practical usages issues, to which extend memorization effect can happen is hard to quantize. Thus, performance obtained from existing solutions are far from desired, and we are motivated to solve this issue by AutoML.

## 2.2 AUTOMATED MACHINE LEARNING (AUTOML)

Automated machine learning (AutoML) (Hutter et al., 2018) has recently exhibited its power in easing the usage of and designing better machine learning models. Basically, AutoML can be regarded as a black-box optimization problem where we need to efficiently and effectively search for hyper-parameters or designs for the underlying learning models evaluated by the validation set. Regarding the success of AutoML, there are two important perspectives (Feurer et al., 2015; Zoph & Le, 2017; Xie & Yuille, 2017; Bender et al., 2018):

- *Search space*: First, it needs to be general enough, which means it should cover existing models as special cases. This also helps experts better understand limitations of existing models and thus facilitate future researches. However, the space cannot be too general, otherwise searching in such a space will be too expensive.
- *Search algorithm*: Optimization problems in AutoML are usually black-box. Unlike convex optimization, there is no universal and efficient optimization tools. Once the search space is determined, domain knowledge should also be explored in the design of search algorithm so that good candidates in the space can be identified efficiently.

Search space is domain-specific and needs to specially designed for every AutoML problem. There are two types of search algorithms popularly used. The first one is derivative-free optimization methods, it is usually used for searching in a general search space, e.g., reinforcement learning

(Zoph & Le, 2017; Baker et al., 2017), genetic programming (Escalante et al., 2009; Xie & Yuille, 2017), and Bayes optimization (Feurer et al., 2015; Snoek et al., 2012). More recently, gradient-based methods, which alternatively update parameters and hyper-parameters, have been developed as more efficient replacements for derivative-free optimization methods on some AutoML problems, e.g., neural network architecture search (Liu et al., 2019; Akimoto et al., 2019; Xie et al., 2018).

However, existing AutoML techniques cannot be directly used in exploiting memorization here. First, we need to carefully define a domain-specific space. Besides, we will also show existing algorithms are neither not applicable nor too slow. This motivates us to propose a new algorithm based on natural gradient.

## 3 THE PROPOSED METHOD

Here, we first give a closer look at why it is difficult to exploit the memorization effect (Sec.3.1). This also helps us identify key observations on how memorization can happen. This observation subsequently enables us to design an expressive but compact search space (Sec.3.2), and motivates us to use natural gradient method (Amari, 1998; Ollivier et al., 2017) that can generate gradients in the parameterized space for efficient optimization (Sec.3.3).

### 3.1 DIFFICULTIES IN EXPLOITING MEMORIZATION

As in Sec.2.2, an important challenge in designing search spaces is to balance the size (or dimension) and the expressive ability of the search space. An overly constrained search space may not contain candidates that have a satisfying performance, whereas too large search spaces will be difficult to effectively search. All these require us to have an in-depth understanding of the difficult in designing of $R(T)$, which is at step 8 of Alg.1 and is used to exploit the memorization effect. Thus, we are motivated to look at factors which can affect the memorization of deep networks, and to seek patterns from the resultant influences. Specifically, we examine memorization when data sets, architectures, or optimizers are changed. Results are in Fig.1. From these figures, we can observe:

- *Foot-stone of space design*: There exists a general pattern among all cases, i.e., all models' test accuracy will first increase, then decrease.

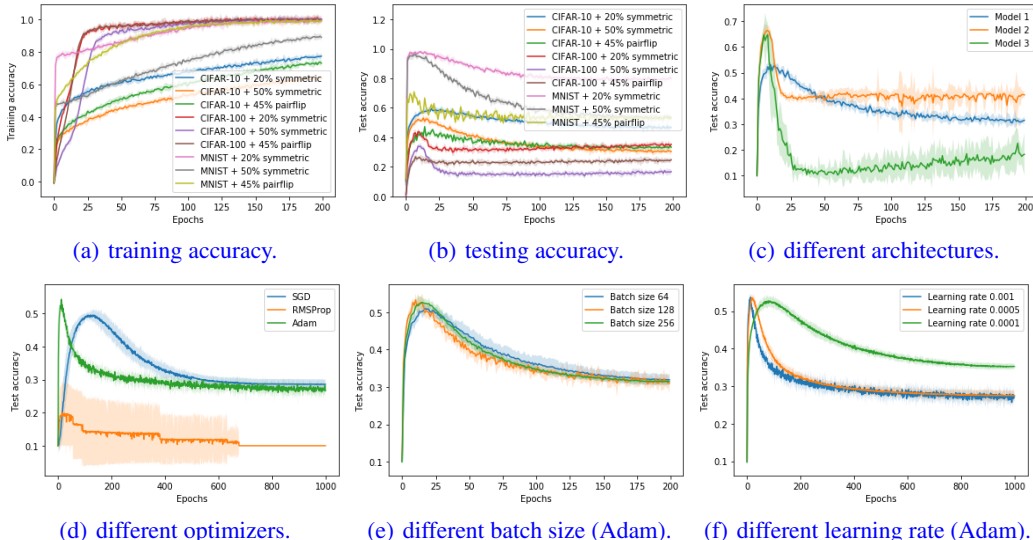

(a) training accuracy. (b) testing accuracy. (c) different architectures.

(d) different optimizers. (e) different batch size (Adam). (f) different learning rate (Adam).

Figure 1: Memorization effects are shown in Fig.(a) and (b), where training accuracy continuously increases while testing accuracy first increases and then significantly decreases due to noisy labels. Fig.(c-f) show memorization not only heavily depends on data sets, and also many factors (details of experiment are in Appendix A.1).

This general pattern is consistent with that in the literature (Zhang et al., 2016; Arpit et al., 2017; Tanaka et al., 2018; Han et al., 2018b). It is also the key domain knowledge for designing an

expressive and compact search space, which makes the subsequent search possible. However, the more important observation is:

- *Need of AutoML*: Curvature can be significantly affected by these factors. When the peak will appear (i.e., stop learning from simple patterns and start to over-fit), and to which extend the performance will drop from peak (i.e., over-fit on noisy labels) are all hard to quantize.

This observation shows a great variety in appearances of the memorization effect, which further poses a significant need for automated exploiting of such effect. Besides, it is hard to know in advance what learning curve will exactly look like in advance and thus impossible to manually design before learning is performed.

## 3.2 SEARCH SPACE

Recall that in step 8 of Alg.1, $R(t)$ controls how many instances are kept in each mini-batch, and we want to exploit the memorization effect through the variation of $R(t)$. Based on the first empirical observation in Sec.3.1, our design of $R(t)$ first should satisfy:

- *Curvature*: $R(t)$ should be similar with the inverse of the learning curve. In other words, $R(t)$ should first drop, then (possibly) rise.

The reason behind this constraint is as follows: Since the learning curve represents the model's accuracy, when it rises, we should drop more large-loss samples as the large loss is more likely the result of noisy labels than model's misclassification. And when the learning curve falls, we should drop less to help the model learn more.

- *Range*: $R(t) \in [0,1]$ for $t \in \{1, \ldots, T\}$ with $R(1) = 1$.

Since $R(t)$ denotes the proportion of selected instances, it is naturally in $[0,1]$. Besides, at the beginning, dropping small-loss samples will be same as dropping samples randomly, and we need to pass sufficient number of instances such that the model can learn from patterns.

|  | functions | definition |
|---|---|---|
| decreasing | power (pow-d) | $1/(1+bt)^a$ |
|  | exponential (exp-d) | $e^{-bt^a}$ |
|  | logarithmic (log-d) | $\log(b)/\log(at+b)$ |
| increasing | power (pow-i) | $b(t/T)^a$ |
|  | logarithmic (log-i) | $b\frac{\log(1+t^a)}{\log(1+T^a)}$ |

Figure 2: Construction elements for $f_i$ (cf. left table) and some approximated curves as examples (cf. right figure). Target is an example of the ground-truth $R(t)$ to approximate.

As $R(t)$ itself can be seen as a function which takes $t$ as input and outputs a scalar, we can use some simple $f_i$ as basis functions to approximate the complicate $R(t)$. We list some simple monotonously decreasing or increasing functions in Fig.2 and express $R(t)$ as a linear combination of them:

$$R(t) = \sum_{i=1}^{k} f_i(t; \alpha_i), \text{ s.t. } \boldsymbol{\alpha} \in \mathcal{A}, \tag{1}$$

where the $\alpha_i$ is the hyper-parameters controlling each term (for example $a$ and $b$ in Fig.2).

Let the clean validation set be $\mathcal{G}_{val}$, $F$ be Alg.1 with network parameter $\boldsymbol{w}$ and sampling rule $\boldsymbol{\alpha}$ ($R(t)$ is parameterized by Eqa.(1)). $\mathcal{M}(\boldsymbol{w}, \mathcal{G}_{val})$ measures the validation performance with the model parameter $\boldsymbol{w}$. Thus, memorization effect can be automatically exploited by solving the following problem

$$\boldsymbol{\alpha}^* = \arg\max_{\boldsymbol{\alpha} \in \mathcal{A}} \mathcal{M}\left(\boldsymbol{w}^*(\boldsymbol{\alpha}), \mathcal{G}_{val}\right), \text{ s.t. } \boldsymbol{w}^*(\boldsymbol{\alpha}) = \arg\min_{\boldsymbol{w}} F(\boldsymbol{w}; \boldsymbol{\alpha}). \tag{2}$$

Then, the optimal curvature is derived from $R(t) = \sum_{i=1}^{k} f_i(t; \alpha_i^*)$.

**Remark 3.1.** *In Co-teaching algorithm (Han et al., 2018b), $R(t)$ is determined as follows*

$$R(t) = 1 - \tau \cdot \min \left( (t/t_k)^c, 1 \right), \tag{3}$$

*which contains three hyper-parameters: $\tau$, $c$ and $t_k$. Thus, we can formulate a search space as: $\tau \in (0,1), c \in (0,+\infty)$ and $t_k \in \{1, \cdots, T\}$. The expressivity of such $R(t)$ is not enough as Eqa.(1) here (since Eqa.(3) is covered as a special case).*

## 3.3 SEARCH ALGORITHM

Here, we first discuss problems of using gradient-based methods here (Sec.3.3.1). These problems motivate us design an efficient search algorithm based on natural gradient algorithm (Sec.3.3.2).

### 3.3.1 ISSUES OF EXISTING ALGORITHMS

Gradient-based methods need the chain rule to obtain the gradient w.r.t hyper-parameters from the gradient w.r.t network weights, i.e., $\nabla_{\boldsymbol{\alpha}} \mathcal{M} = \partial \mathcal{M}/\partial \boldsymbol{w} \cdot \partial \boldsymbol{w}/\partial \boldsymbol{\alpha}$. This condition does not hold for our problem since our hyper-parameters control *how many* samples will be used to update the weights. Thus, it is hard to compute $\nabla_{\boldsymbol{\alpha}} \mathcal{M}$ here. Besides, the learned parameters $\boldsymbol{w}$ cannot be shared among different $\boldsymbol{\alpha}$. In previous methods, hyper-parameters are not coupled with the training process, e.g., network architectures (Liu et al., 2019; Akimoto et al., 2019) and regularization (Luketina et al., 2016). However, $R(t)$ here will heavily influence the training process, which is also shown in our Fig.1. Thus, gradient-based methods cannot be applied here.

### 3.3.2 PROPOSED ALGORITHM

The analysis above demonstrates that only derivative-free methods are applicable to our problem, which can be slow. Here, we discover that natural gradient (NG) (Amari, 1998; Pascanu & Bengio, 2013; Ollivier et al., 2017), can be used. The most interesting point here is that NG can still benefit from gradient descent, but without the computation of $\partial \mathcal{M}/\partial \boldsymbol{\alpha}$ or sharing of $\boldsymbol{w}$.

The basic idea of NG is summarized as follows: instead of directly optimizing w.r.t $\boldsymbol{\alpha}$, we consider a random distribution $p_\theta(\boldsymbol{\alpha})$ over $\boldsymbol{\alpha}$ parametrized by $\theta$, and maximize the expected value of our validation performance $\mathcal{M}$ w.r.t $\theta$, i.e.,

$$\max_{\boldsymbol{\theta}} \mathcal{J}(\boldsymbol{\theta}) \equiv \int_{\boldsymbol{\alpha} \in \mathcal{A}} \mathcal{M} \left( \boldsymbol{w}^*(\boldsymbol{\alpha}), \mathcal{G}_{val} \right) p_{\boldsymbol{\theta}}(\boldsymbol{\alpha}) \, d\boldsymbol{\alpha}, \text{ s.t. } \boldsymbol{w}^*(\boldsymbol{\alpha}) = \arg \min_{\boldsymbol{w}} F(\boldsymbol{w}; \boldsymbol{\alpha}). \tag{4}$$

To optimize $\mathcal{J}(\boldsymbol{\theta})$ w.r.t $\boldsymbol{\theta}$, NG updates $\boldsymbol{\theta}$ by

$$\boldsymbol{\theta}^{m+1} = \boldsymbol{\theta}^m + \rho \boldsymbol{H}^{-1}(\boldsymbol{\theta}^m) \nabla_{\boldsymbol{\theta}} \mathcal{J}(\boldsymbol{\theta}^m), \tag{5}$$

where $\rho$ is the step-size, $\boldsymbol{H}(\boldsymbol{\theta}^m)$ is the Fisher information matrix at $\boldsymbol{\theta}^m$, and

$$\nabla_{\boldsymbol{\theta}} \mathcal{J}(\boldsymbol{\theta}) = \int_{\boldsymbol{\alpha} \in \mathcal{A}} \mathcal{M} \left( \boldsymbol{w}^*(\boldsymbol{\alpha}), \mathcal{G}_{val} \right) \nabla_{\boldsymbol{\theta}} p_{\boldsymbol{\theta}}(\boldsymbol{\alpha}) d\boldsymbol{\alpha} = \mathbb{E}_{p_{\boldsymbol{\theta}}(\boldsymbol{\alpha})} \left[ \mathcal{M} \left( \boldsymbol{w}^*(\boldsymbol{\alpha}), \mathcal{G}_{val} \right) \nabla_{\boldsymbol{\theta}} \log p_{\boldsymbol{\theta}}(\boldsymbol{\alpha}) \right]. \tag{6}$$

By sampling candidate $R(t)$s from the given distribution $p_{\boldsymbol{\theta}^m}$ in each iteration, Eqa.(6) just needs validation's performance (without computation of $\partial \mathcal{M}/\partial \boldsymbol{\alpha}$). Besides, such gradient has proved to give the steepest ascend direction in the probabilistic space (Theorem 1 in (Angluin & Laird, 1988)).

The last question is how to choose $p_{\boldsymbol{\theta}}$, which may significantly influence the algorithm's convergence behavior. Fortunately, NG exhibits strong robustness against different choices over $p_{\boldsymbol{\theta}}$. The reason is that Fisher matrix, which encodes the second order approximation to the Kullback-Leibler divergence, can accurately capture curvatures introduced by various $p_{\boldsymbol{\theta}}$. Thus, NG descent is parameterization-invariant, has good generalization ability, and moreover can be regarded as a second-order method in the space of $\boldsymbol{\theta}$ (Ollivier et al., 2017). The proposed search algorithm is in Alg.2 (details are in Appendix A.2). As will be shown in experiments, all these properties make NG an ideal search algorithm here, and can be faster than other popular AutoML approaches.

## 4 EXPERIMENTS

We implement all our experiments using PyTorch 0.4.1 on a GTX 1080 Ti GPU.

---

**Algorithm 2** Proposed search algorithm (based on natural gradient).

---

1: **while** not converged **do**
2:    **for** $n = 1, \cdots, N$ **do**
3:       draw an $\boldsymbol{\alpha}$ from the current distribution $p_{\boldsymbol{\theta}}(\boldsymbol{\alpha})$;    // *approximate gradients*
4:       run Alg.1 using $R(t)$ described by Eqa.(1);    // *no weight-sharing*
5:    **end for**
6:    use samples in step 2-5 to compute Fisher information matrix;
7:    update $\boldsymbol{\theta}$ by Eqa.(5) and (6);
8: **end while**

---

### 4.1 EXPERIMENTS ON SYNTHETIC DATA

In this section, we demonstrate the superiority of the proposed search space and search algorithm on the synthetic data. The ground-truth $R(t)$ is shown in Fig.3(a), which is an example curvature satisfying two requirements in Sec.3.2.

#### 4.1.1 SEARCH SPACE COMPARISON

The goal here is to approximate $R(t)$ in Fig.3(a). Let the estimated function be $\bar{R}(t)$ where $t = 1, \cdots, 200$. The target is to minimize RMSE, i.e., $f(\bar{R}) = [1/200 \sum_{t=1}^{200} (R(t) - \bar{R}(t))^2]^{1/2}$. Three different search spaces are compared: 1). *Full space*: $\bar{R}_1(t) = \{e_1, \cdots, e_{200}\}$, i.e., there is no constraint in the space, and estimation for $R(t)$ at each time stamp is performed independently; 2). *Co-teaching*'s space in Eqa.(3), i.e., $R_2(t) = 1 - \tau \min((t/t_k)^c, 1)$, and $\{\tau, c, t_k\}$ needs to be estimated; and 3). The *proposed* space in Eqa.(1), which encodes our observations in Sec.3.1. Random search (Bergstra & Bengio, 2012) is performed in all three spaces.

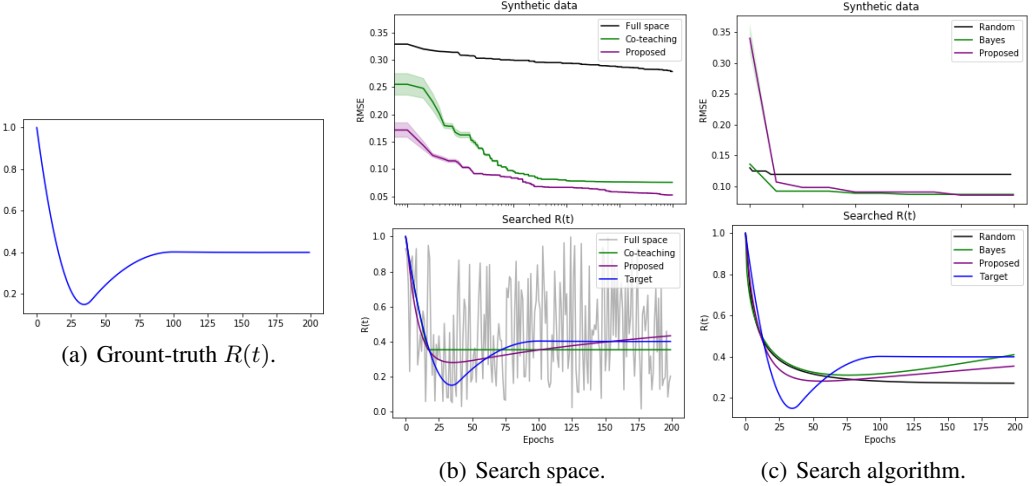

(a) Grount-truth $R(t)$.          (b) Search space.          (c) Search algorithm.

Figure 3: Comparison of the search space (cf. the middle figure) and search algorithm (cf. the right figure) on the synthetic data.

Results are in Fig.3(b). We can see that no constraints on the search space will lead to disastrous performance and extremely slow convergence rate. And compare our space with previous Co-teaching's space, we can see that our space can approximate the target better due to its larger degree of freedom.

#### 4.1.2 SEARCH ALGORITHM COMPARISON

We first test our proposed method's efficiency over other hyper-parameter optimization (HPO) algorithms on a synthetic problem as follows: Given a pre-defined $R(t)$ as target, we try to search for a $R(t)$ that has the lowest squared loss to the target. We compare our proposed method with *random search* (Bergstra & Bengio, 2012) and *Bayesian optimization* (Kandasamy et al., 2019), which are two popular methods for hyper-parameter optimization. Results are in Fig.3(c). The results demonstrate that our proposed natural gradient approach has the fastest convergence rate. And it can find the best $R(t)$ with the least RMSE to the target.

## 4.2 EXPERIMENTS ON BENCHMARK DATA SETS

We verify the efficiency of our approach on three benchmark data sets, i.e., MNIST, CIFAR-10 and CIFAR-100 (details are in Appendix A.3). These data sets are popularly used for the evaluation of learning with noisy labels in the literature (Zhang et al., 2016; Arpit et al., 2017; Jiang et al., 2018; Han et al., 2018b). Following (Patrini et al., 2017; Han et al., 2018b; Chen et al., 2019), we corrupt these data sets manually by two types of noise. (1) Symmetry flipping (with 20% and 50% noise level) and (2) Pair flipping (with 45% noise level).

All these noise patterns correspond to real-world scenarios. For example, on the macro-level, class cat flipping to the class dog makes sense, while class dog flipping to class cat also makes sense. Such flipping yields a noise pattern called symmetric-flip (Patrini et al., 2017). On the micro-level, for dogs, class Norfolk terrier flipping to class Norwich terrier makes sense, while class Norfolk terrier flipping to class Australian terrier not. This flipping yields a noise pattern called pair-flip (Han et al., 2018a), which depicts the fine-grained classification case.

We set the network architecture as the same in (Yu et al., 2019) (Appendix A.4). To measure the performance, same as (Patrini et al., 2017; Han et al., 2018b; Chen et al., 2019), we use the test accuracy, i.e., $test\ accuracy = (\#of\ correct\ predictions)/(\#of\ test\ dataset)$. Specifically, the full learning curve, i.e., testing accuracy v.s. epochs, is reported. Ideally, if a method is robust to noisy labels, its performance will increase with more training epochs. Thus, if the learning curve of one method quick falls down after reaching the maximum, then it means the method is not robust intrinsically (Zhang et al., 2016; Arpit et al., 2017).

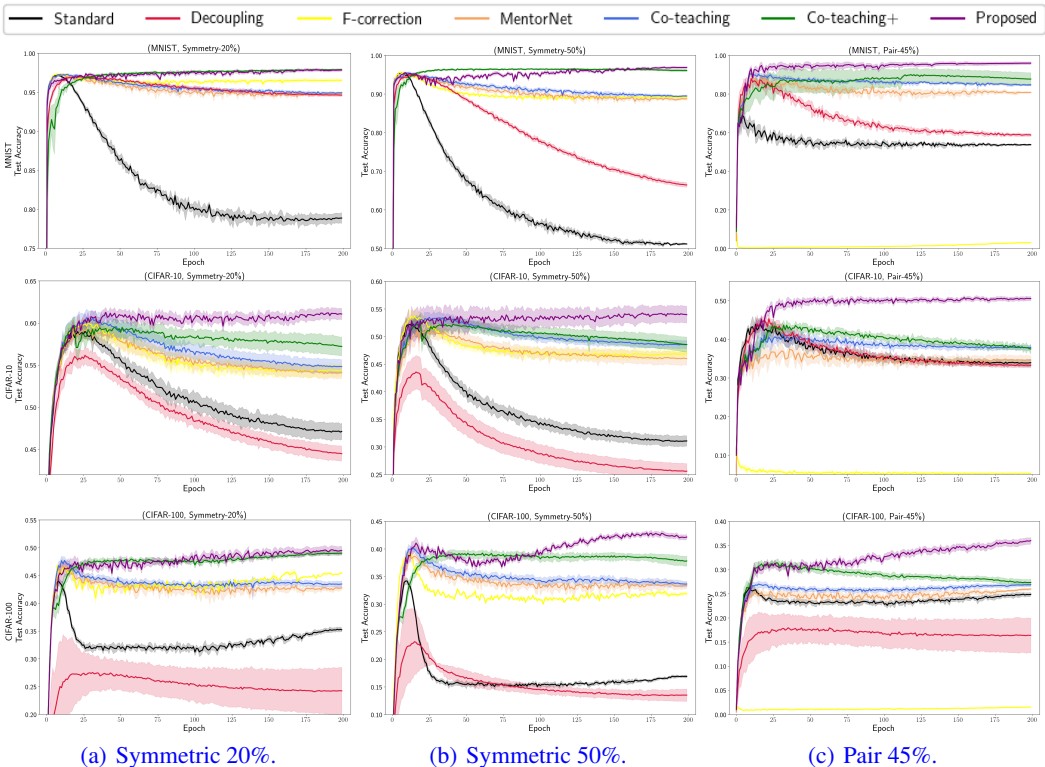

(a) Symmetric 20%.      (b) Symmetric 50%.      (c) Pair 45%.

Figure 4: Comparison on testing accuracy between the proposed and other human-designed methods. Top to bottom: MNIST, CIFAR-10 and CIFAR-100. Best testing accuracy of each method is reported in Appendix B.1.

### 4.2.1 COMPARISON ON LEARNING PERFORMANCE

To show better accuracy can be achieved by automatically exploiting the memorization effect, we compared the proposed method with 1). *MentorNet* (Jiang et al., 2018); 2). *Co-teaching* (Han et al., 2018b); 3). *Co-teaching+* (Yu et al., 2019) (enhancing Co-teaching by disagreements on

predictions); 4). *Decoupling* (Malach & Shalev-Shwartz, 2017); 5). *F-correction* (Patrini et al., 2017); 6). As a simple baseline, we also compare with the standard deep network that directly trains on noisy datasets (abbreviated as *Standard*). As an example usage, the proposed method is combined with Co-teaching, i.e., Co-teaching is run with search $R(t)$. Fig.4 shows the comparison with various human-designed methods. We can see that the proposed method significantly outperforms existing methods by a large margin, especially on the more noisy cases (i.e., symmetric-50% and pair-45%).

Besides, the proposed method not only beats Co-teaching due to better exploiting of the memorization effect, but also wins Co-teaching+, which further filter small-loss instances in Co-teaching by checking disagreements on predictions of labels. These demonstrates the importance and benefits of searching proper $R(t)$.

### 4.2.2  CASE STUDY ON SAMPLING RATE

To understand why the proposed method can obtain much higher testing accuracy, we plot the searched $R(t)$ in Fig.5. We can see that all methods finally drop more large-loss instances than actual noise level. The reason is very intuitive, a large-loss instance usually also has larger gradient, and it can have much more influences than several clean small-loss instances if its label is wrong. Thus, we may want to drop more samples eventually. However, this is not an easy task, as in Tab.8 of (Han et al., 2018b), simply making $\tau$ larger sometimes leads to decrease in testing accuracy; and in the third row of Fig.5, we may also need to keep more samples during the training. A more evolved examination on precision of clean labels are in App.B.3.

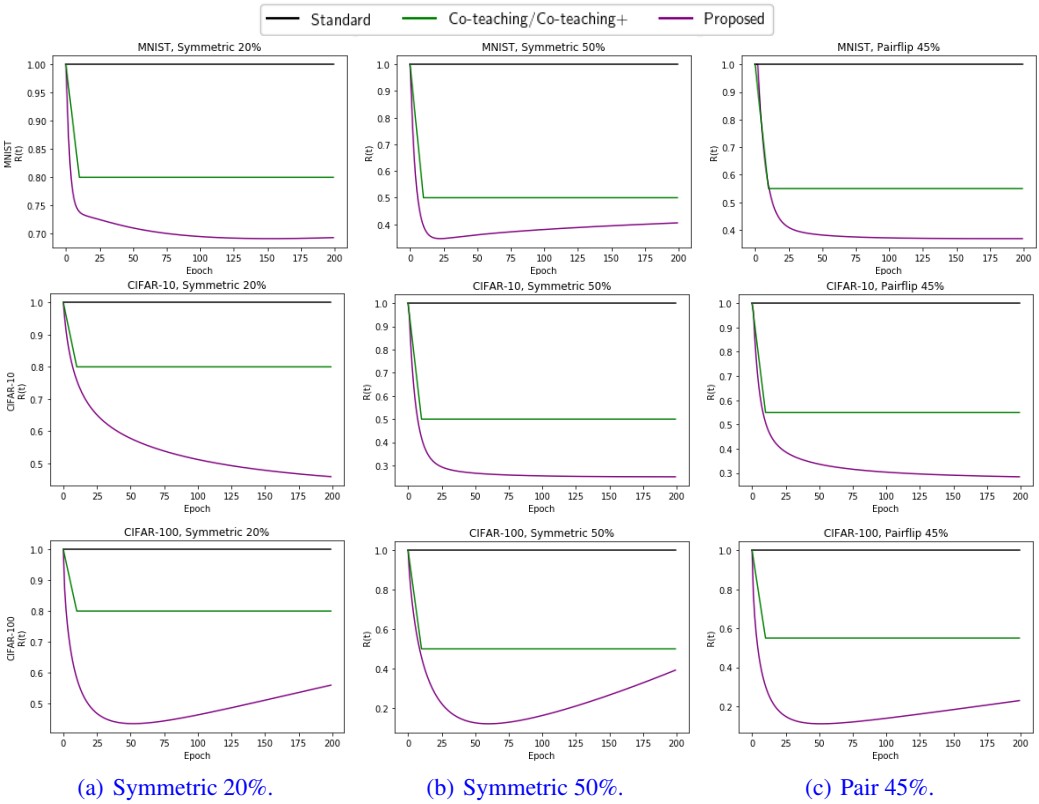

(a) Symmetric 20%.        (b) Symmetric 50%.        (c) Pair 45%.

Figure 5: Comparison on $R(t)$ between the proposed (automatically searched from the data) and Co-teaching (note that Co-teaching+ uses the same $R(t)$ as Co-teaching). Top to bottom: MNIST, CIFAR-10 and CIFAR-100.

### 4.2.3  COMPARISON WITH HPO METHODS

Finally, in this section, we compare the proposed natural gradient (NG) algorithm with 1). *random search* (Bergstra & Bengio, 2012) and 2). *Hyperband* (Li et al., 2017a). Note that Bayesian optimization is slower than *Hyperband*, thus not compared (*Hyperband* cannot be used in Sec.4.1.2

due to no inner loops). Besides, reinforcement learning (RL) (Zoph & Le, 2017) is not compared as the searching problem is not a multi-step one. Genetic programming (Xie & Yuille, 2017) is not considered neither, as the search space is a continuous one. Fig.6 compares the proposed method with random search and Hyperband. From the figure, we can see that natural gradient converges faster than other HPO methods in this problem. Our proposed method can also find better $R(t)$s under different data sets and noise settings.

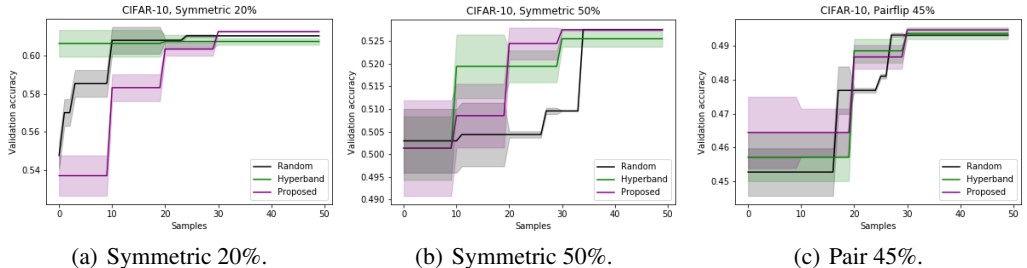

| (a) Symmetric 20%. | (b) Symmetric 50%. | (c) Pair 45%. |
| --- | --- | --- |

Figure 6: Comparison between the proposed search and other HPO algorithm. CIFAR-10 is used.

## 4.3 EXPERIMENTS ON REAL FACE DATA SET

Finally, we test the validity of our approach on real data sets. We conduct experiments on the application of deep face recognition (Parkhi et al., 2015). Following (Wang et al., 2019), we use VggFace2-R (Cao et al., 2018) data set, which is a noisy data set collected from Google image search, for training, then tune hyper-parameters and report testing accuracy on four small and clean image data sets, i.e., CALFW (Zheng et al., 2017), CPLFW (Zheng & Deng, 2018), AgeDB (Moschoglou et al., 2017), and CFP (Sengupta et al., 2016) (details are in Appendix A.3).

We compare the performance of our proposed method with those methods used in Sec.4.2.1 and Co-mining (Wang et al., 2019). Testing accuracy is reported in Tab.1. We can see that, our proposed method consistently achieves the best performance. Thus, our method is not only useful with synthetic noise but also work well on real applications.

Table 1: Experiments on deep face recognition with real data sets. All models are trained with noisy VggFace2-R data set, and evaluate on the four clean test data sets. Except the proposed method, all performance are copied from (Wang et al., 2019); F-correction is not reported as its performance is not available in (Wang et al., 2019).

|  | CALFW | CPLFW | AgeDB | CFP | average |
| --- | --- | --- | --- | --- | --- |
| Standard | 90.11 | 86.30 | 92.81 | 95.50 | 91.18 |
| Decoupling | 90.23 | 86.14 | 93.90 | 95.85 | 91.53 |
| MentorNet | 90.14 | 85.41 | 92.70 | 95.20 | 90.86 |
| Co-teaching | 89.90 | 85.05 | 92.05 | 95.05 | 90.62 |
| Co-teaching+ | 89.43 | 85.23 | 92.50 | 95.41 | 90.64 |
| Co-Mining | 91.06 | 87.31 | 94.05 | 95.87 | 92.07 |
| Proposed | **92.04** | **89.43** | **95.22** | **96.16** | **93.20** |

## 5 CONCLUSION

In this paper, motivated by the main difficulty that to what extent the memorization effect of deep networks can happen, we propose to exploit memorization by automated machine learning (AutoML) techniques. This is done by first designing an expressive but compact search space, which is based on observed general patterns for memorization, and designing a natural gradient-based search algorithm, which overcomes the problem of non-differential and failure of parameter-sharing. Extensive experiments on both synthetic data sets and benchmark data sets demonstrate that the proposed method can not only be much efficient than existing AutoML algorithms, but also achieve much better performance than the state-of-the-art sample-selection approach.

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

# A   IMPLEMENTATION DETAILS

## A.1   DETAILS FOR FIG.1

(a-b) We use the same MNIST/CIFAR-10/100 datasets and their details are in Appendix A.3. The network model is model 1 and its details are given in Appendix A.4. The number of training epochs is 200 and the batch size is set to 128. The initial learning rate for the Adam optimizer is 0.001 and it is linearly decayed to zero from the 80th epoch to final.

(c) Details of each model is listed in Appendix A.4. Note that model 2 on CIFAR-10 has the output dimension of 10 instead of 100. The number of training epochs, batch size, optimizer and learning rate schedule follows (a) and (b).

(d) The learning rate schedule for Adam follows (a) and (b). The initial learning rate for the SGD optimizer is 0.1, and decay to 0.01 and 0.001 from the 500th and 750th epoch, respectively. And for the RMSProp optimizer, the learning rate is set constantly to 0.01. To demonstrate the memorization effects for SGD optimizer, the number of training epochs is 1000 instead of 200. The model is model 1 and the batch size is 128.

(e) The model, number of training epochs, optimizer and learning rate schedule is the same as (a) and (b). In a word, we only change the batch size.

(f) We only change the initial learning rate for the learning rate schedule. Also, to demonstrate the memorization effects for small learning rate cases, the number of training epochs is 1000 instead of 200. And the model, batch size, optimizer all follow (a) and (b).

## A.2   NATURAL GRADIENT

Since our search space for $\alpha$ is bounded, we use Beta distribution in natural gradients, i.e. $p_\theta(\lambda) = \prod_{i=1}^{k} Beta(\alpha_i; \theta_i)$. Each Beta distribution has two parameters $x, y$ and the probability density function (PDF) is given by

$$Beta(\alpha; x, y) = \Gamma(x+y)/\Gamma(x)+\Gamma(y) \alpha^{x-1}(1-\alpha)^{y-1},$$

where $\Gamma(x)$ is the gamma function. The Fisher information matrix

$$\boldsymbol{H}(\theta) = \mathbb{E}_{p_\theta(\boldsymbol{\alpha})} \left[ \nabla_\theta \log p_\theta(\boldsymbol{\alpha}) \nabla_\theta \log p_\theta(\boldsymbol{\alpha})^T \right],$$

is estimated by samples since no explicit formula is available. And the gradient in Fisher $\nabla_\theta \log p_\theta(\boldsymbol{\alpha})$ is different for different parameters $x$ and $y$, as

$$\nabla_x \log Beta(\alpha; x, y) = \log(\alpha) + \psi(x + y) - \psi(x) \text{ and}$$
$$\nabla_y \log Beta(\alpha; x, y) = \log(1 - \alpha) + \psi(x + y) - \psi(y),$$

where $\psi(x)$ denotes the digamma function.

For synthetic data, we run for 10 iterations and in each iteration, we sample 10 samples and evaluate its performance. The learning rate for natural gradient is set to 80.

For benchmark and real noisy data sets, all search algorithms will run for 5 iterations in total. In random search and natural gradient, we sample 10 samples for CIFAR-10 and 20 samples for other four data sets to evaluate its performance in each iteration. The learning rate for natural gradient is set to 20, 500, 80, 30 for MNIST, CIFAR-10, CIFAR-100 and VggFace2-R data set respectively.

To obtain a better estimation of the Fisher information matrix, in spite of those samples for evaluation, we also sample another 10000 samples to compute the Fisher in each iteration.

## A.3   DATA SETS

We obtain MNIST, CIFAR-10, CIFAR-100 from PyTorch's torchvision package, and VggFace2-R, CALFW, CPLFW, AgeDB, CFP from its original source. Some statistics about those datasets are given below.

Table 2: Summary of data sets used in the experiments.

| | # of train | # of validation | # of test | # of class |
|---|---|---|---|---|
| MNIST | 60,000 | 5,000 | 5,000 | 10 |
| CIFAR-10 | 50,000 | 5,000 | 5,000 | 10 |
| CIFAR-100 | 50,000 | 5,000 | 5,000 | 100 |
| VggFace2-R | 3.31M | (Train only) | (Train only) | 9,131 |
| CALFW | (Validation & test only) | 6,087 | 6,087 | 5,749 |
| CPLFW | (Validation & test only) | 5,826 | 5,826 | 5,749 |
| AgeDB | (Validation & test only) | 8,244 | 8,244 | 568 |
| CFP | (Validation & test only) | 3,500 | 3,500 | 500 |

## A.4 NETWORK STRUCTURE

For VggFace2-R, we use ResNet-50 as in (Wang et al., 2019). Adam optimizer (Kingma & Ba, 2014) (momentum=0.9) is used with an initial learning rate of 0.001, and the batch size is set to 128 and we run 200 epochs. The learning rate is linearly decayed to zero from 80 to 200 epochs.

Table 3: MLP and CNN models used in our experiments.

| MLP on MNIST | CNN on CIFAR-10 (Model 1) | CNN on CIFAR-100 (Model 2) | Model 3 |
|---|---|---|---|
| 28×28 Gray Image | 32×32 RGB Image | 32×32 RGB Image | 32×32 RGB Image |
| Dense 28×28→256 ReLU | 5×5 Conv, 6 ReLU 2×2 Max-pool | 3×3 Conv, 64 BN, ReLU 3×3 Conv, 64 BN, ReLU 2×2 Max-pool | 3×3 Conv, 128 BN, LReLU 3×3 Conv, 128 BN, LReLU 3×3 Conv, 128 BN, LReLU 2×2 Max-pool, stride 2 Dropout, p=0.25 |
| | 5×5 Conv, 16 ReLU 2×2 Max-pool | 3×3 Conv, 128 BN, ReLU 3×3 Conv, 128 BN, ReLU 2×2 Max-pool | 3×3 Conv, 256 BN, LReLU 3×3 Conv, 256 BN, LReLU 3×3 Conv, 256 BN, LReLU 2×2 Max-pool, stride 2 Dropout, p=0.25 |
| | Dense 16×5×5→120 ReLU Dense 120→84 ReLU | 3×3 Conv, 196 BN, ReLU 3×3 Conv, 196 BN, ReLU 2×2 Max-pool | 3×3 Conv, 512 BN, LReLU 3×3 Conv, 256 BN, LReLU 3×3 Conv, 128 BN, LReLU Avg-pool |
| Dense 256→10 | Dense 84→10 | Dense 256→100 | Dense 128→ 10 |

# B   MORE RESULTS

## B.1   PERFORMANCE COMPARISON WITH EARLY STOPPING

To make a better comparison, we apply early-stop trick to all those methods in Sec.4.2 and report their best accuracy instead of the final accuracy. The accuracy is averaged in 5 runs and we also report the standard deviation. Results are shown below:

Table 4: Testing accuracy (in percentage) with early stop on MNIST.

| noise | symmetric 20% | symmetric 50% | pairflip 45% |
|---|---|---|---|
| Standard | 97.16±0.20 | 95.40±0.09 | 68.86±5.20 |
| Decoupling | 97.09±0.12 | 94.88±0.39 | 88.03±1.76 |
| F-correction | 97.32±0.13 | 95.59±0.30 | 9.50±1.55 |
| MentorNet | 97.23±0.11 | 95.32±0.23 | 90.47±2.09 |
| Co-teaching | 97.26±0.14 | 95.42±0.27 | 90.49±1.52 |
| Co-teaching+ | 97.88±0.10 | 96.52±0.10 | 90.00±3.16 |
| Proposed | **97.89±0.21** | **96.91±0.25** | **95.94±0.70** |

Table 5: Testing accuracy (in percentage) with early stop on CIFAR-10.

| noise | symmetric 20% | symmetric 50% | pairflip 45% |
|---|---|---|---|
| Standard | 59.04±0.96 | 52.23±1.32 | 44.01±1.49 |
| Decoupling | 56.16±1.02 | 43.57±3.03 | 45.31±1.00 |
| F-correction | 59.94±0.77 | 53.71±1.48 | 10.22±1.49 |
| MentorNet | 59.34±0.93 | 51.08±1.06 | 37.45±2.45 |
| Co-teaching | 60.62±1.03 | 53.48±0.86 | 41.26±0.74 |
| Co-teaching+ | 59.70±1.07 | 52.49±1.52 | 43.66±1.28 |
| Proposed | **61.27±0.59** | **54.26±1.55** | **50.80±0.55** |

Table 6: Testing accuracy (in percentage) with early stop on CIFAR-100.

| noise | symmetric 20% | symmetric 50% | pairflip 45% |
|---|---|---|---|
| Standard | 43.98±0.87 | 34.19±0.52 | 26.30±0.87 |
| Decoupling | 27.53±3.24 | 23.18±5.85 | 17.95±2.86 |
| F-correction | 46.80±0.13 | 39.20±0.30 | 1.58±0.31 |
| MentorNet | 46.82±1.19 | 38.68±0.39 | 25.98±0.26 |
| Co-teaching | 47.69±0.68 | 40.10±0.54 | 27.01±0.69 |
| Co-teaching+ | 49.03±0.42 | 39.24±0.64 | 31.39±0.89 |
| Proposed | **49.63±0.74** | **42.90±0.40** | **36.00±0.68** |

## B.2   PERFORMANCE COMPARISON WITH SIMPLE DECAY FUNCTION

Here, we compare the performance obtained (under early stop setup in Sec.B.1) from

- the space used in Co-teaching (Han et al., 2018b) (see Remark 3.1);
- a space spanned by a simple decay function (i.e., pow-d in Fig.2); and
- the proposed space in Eqa.(1).

Natural gradient proposed in Sec.3.3.2 is used for optimization. Results are shown in Fig.7. We can see that performance obtained from the proposed method is much better than that from a simple decay function. This again demonstrates the needs of approximating $R(T)$ by a linear combination of some basis functions.

Table 7: Testing accuracy (in percentage) comparison between the proposed method and that from a simple decay function.

|  | noisy | space from Co-teaching | pow-d (Figure 2) | proposed |
|---|---|---|---|---|
| MNIST | symmetric 20% | 97.83±0.70 | 97.67±0.03 | **97.89±0.21** |
|  | symmetric 50% | 96.54±0.40 | 96.56±0.01 | **96.91±0.25** |
|  | pairflip 45% | 93.27±0.80 | 94.99±0.03 | **95.94±0.70** |
| CIFAR-10 | symmetric 20% | 60.72±0.31 | 60.83±0.10 | **61.27±0.59** |
|  | symmetric 50% | 54.11±0.56 | 53.19±0.50 | **54.26±1.55** |
|  | pairflip 45% | 47.48±1.86 | 49.17±0.38 | **50.80±0.55** |
| CIFAR-100 | symmetric 20% | 48.18±1.27 | 47.93±0.02 | **49.63±0.74** |
|  | symmetric 50% | 40.57±0.38 | 41.67±1.57 | **42.90±0.40** |
|  | pairflip 45% | 27.30±0.99 | 32.47±0.05 | **36.00±0.68** |

## B.3 PRECISION OF CLEAN SAMPLES

Here, to understand why the proposed method can lead better performance than Co-teaching, following Han et al. (2018b), we also show label precision, i.e., the average ratio (in percentage) of clean samples in selected samples, in each epoch. Results are in Tab.7. As we can see, precision is consistently much higher than that from Co-teaching. This means the training samples used by the proposed method is cleaner than those used by Co-teaching, and thus better performance can be obtained by the proposed method.

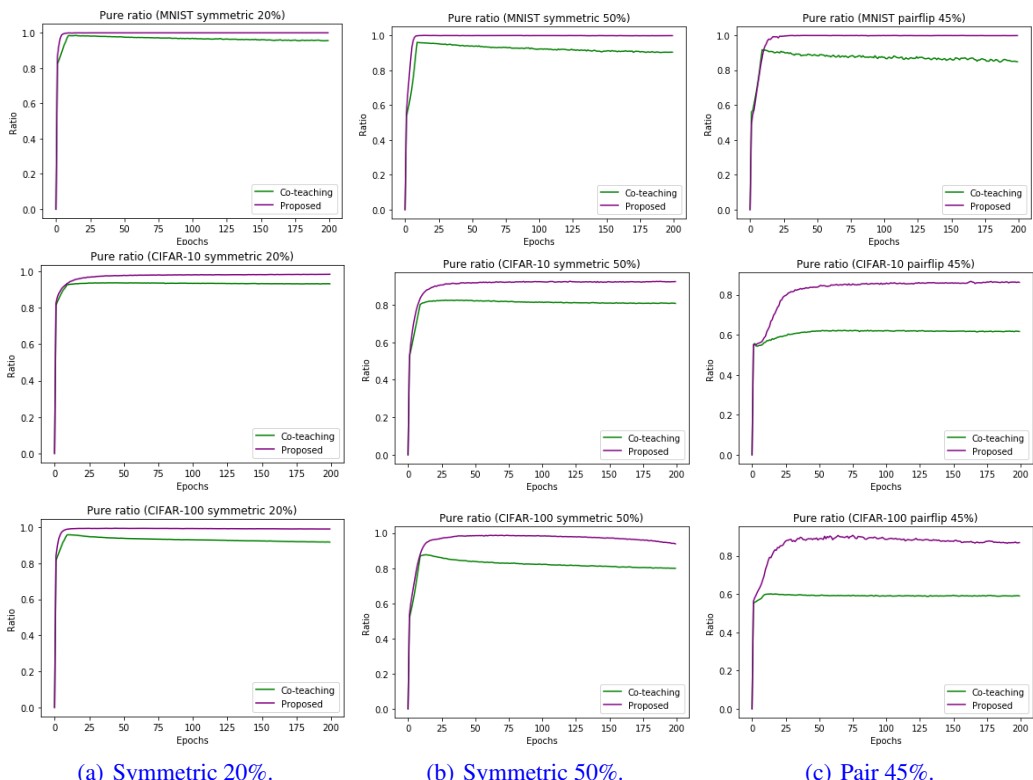

(a) Symmetric 20%.     (b) Symmetric 50%.     (c) Pair 45%.

Figure 7: Comparison on precision of clean samples between the proposed method and Co-teaching. Top to bottom: MNIST, CIFAR-10 and CIFAR-100.

