# OpenReview forum: "Searching to Exploit Memorization Effect in Learning from Corrupted Labels"
_ICLR.cc/2020/Conference — Reject_

### Official Review · AnonReviewer1 · 2019-10-24
**Official Blind Review #3**

**Rating:** 3

**Review:**

This paper studies the problem of learning from corrupted labels via picking up clean instances from training dataset. The sample selection mainly based on function R(t), which controls how many instances are kept. This paper proposes a unique curvature of R(t) based on intuition and presents how R(t) can be learned via combination of some existing functions. Natural gradient is presented to optimize the parameters in the autoML framework. Experimental results on both synthetic data and real-world data demonstrate the effectiveness of the proposed method.

A few comments on this paper:
1. The paper is very verbose and hard to follow. It introduces too many basic concepts in autoML.
2. A key part of the paper is the curvature of R(t), which is based on intuition. Meanwhile, the learned curvature (in Fig 5) doesn't follow the curvature. Does this mean this paper is contradicting its self? The curvature of defined R(t) is not needed?
3. The major difference between this paper and (Han et al. 2018) is how R(t) is defined and learned. The technical contribution of this paper is limited.

Minor comments:
1. For all the figures, it is difficult to view the y-axis (or the y-axis is missing).


**Experience Assessment:**

I have read many papers in this area.

**Review Assessment: Checking Correctness Of Derivations And Theory:**

I assessed the sensibility of the derivations and theory.

**Review Assessment: Checking Correctness Of Experiments:**

I assessed the sensibility of the experiments.

**Review Assessment: Thoroughness In Paper Reading:**

I made a quick assessment of this paper.

---

> ### Author Response · Authors · 2019-11-11
> **Reply to Reviewer#1**
>
> Thanks for your comments. Please note that you are "ICLR 2020 Conference Paper1554 AnonReviewer1".
>
> Q1. It introduces too many basic concepts in autoML
>
> Thanks for the suggestion. In the revised version, we have changed the outline of Section 2.2 and removed unnecessary concepts, e.g., supernet and one-shot. Briefly,
> 1) search space and algorithm are the most two important components in AutoML
> 2) derivative-free and gradient-based are two types of the popular optimization algorithm used
> 3) add a paragraph to clarify the difference between existing AutoML works and the proposed one.
>
> In summary, domain-specific search space and efficient search algorithms are keys to a successful AutoML application (Feurer et al., 2015; Zoph & Le, 2017; Xie & Yuille, 2017; Bender et al., 2018, Hutter et al., 2018).
>
> Q2. The learned curvature (in Fig 5) does not follow the curvature. Does this mean this paper is contradicting itself self?
>
> Please check the revised PDF. The paper is NOT contradicting itself.
> 1) In practice, R(T) correlates with the memorization effect, which heavily depends on many factors (see Figure 1). Thus, "Target" in Figure 2 only represents one possible curvature of R(T), and it does not mean every R(T) should look similar to "Target".
> 2) In the revised version, we have updated Figure 5. The searched R(T) enjoys much diversity, and some look similar to "Target" now, i.e., those on CIFAR-100 (the last row in Figure 5).
>
> Q3. The major difference between this paper and (Han et al. 2018) is how R(t) is defined and learned. The technical contribution of this paper is limited. & The curvature of defined R(t) is not needed?
>
> Thanks for pointing this out. Please see our reply to the Q2 and Q3 for all reviewers. Briefly,
> 1) Identifying that why R(T) is hard to be searched is the first contribution.
> 2) After that, indeed, the difference is only on R(T) compared with Co-teaching. However, how to find a proper R(T) is a non-trivial problem. Please see Figure 1 and 5 in the updated version, R(T) depends on many factors and can exhibit a diverse pattern.
> 3) It is the proposed approach that can boost Co-teaching and then get consistently better performance on synthetic (Section 4.1), benchmark (Section 4.2), and real (Section 4.3) data sets.  Specifically, Co-teaching with searched R(T) can even beat methods that use better criterions to find clean samples, i.e., Proposed v.s. Co-teaching+ (Yu et al., 2019) in Figure 4 and Proposed v.s. Co-Mining (Wang et al., 2019) in Table 1.

---

### Official Review · AnonReviewer3 · 2019-10-26
**Official Blind Review #3**

**Rating:** 3

**Review:**

This paper develops a method for sample selection that exploits the memorization effect. In essence, the authors adopt the co-teaching  (Han et al. NeurIPS 2018) and MentorNet (Jiang et al., ICML 2018) framework, which selects some fraction of examples per minibatch that are hopefully "cleaner" than noisier examples to compute updates from. While in Han et al. the number of instances R selected depends on the number of epochs that have been completed, this paper instead seeks to learn R by approximating it as a linear combination of different types of basis functions  and using natural gradient as the search algorithm. The search space proposed by the authors seem comprehensive: it encompasses the search space of co-teaching, the prior state of the art. Results on synthetic tasks as well as MNIST/CIFAR appear to show the superiority of the proposed method over random search, co-teaching, and other baselines, although the results don't seem conclusive. Overall, I have concerns with some of the contributions, experiments, and presentation, which leaves me at a weak reject.

comments:
- Section 3.1 isn't very compelling to me. Experiments done on just CIFAR with two architectures and optimizers are certainly not sufficient to make any broad claims. I don't think this qualifies as a "contribution" of the paper.
- The paper is difficult to understand, and much of this difficulty stems from poor writing / presentation. The plots depicting experimental results are especially hard to follow.
- I'm a little confused with the setup here. Most practitioners use early stopping to halt training after performance on the validation set drops. As such, why should we care about the held-out curve after the maximum is reached? Shouldn't we care more about the training curve, as at some point during training the noisy labels will also be memorized? Isn't this the definition of the "memorization effect"?
- What about standard baseline methods e.g., active learning to help with this problem? Active learning seems highly relevant yet is not mentioned anywhere in this paper.
- Are all of the basis functions in Fig 2 necessary for the performance of the proposed method? How were they selected? Why is this motivated by the Taylor expansion?
- Figure 5 shows a bunch of R(t) curves learned by the proposed approach across a variety of datasets / noise levels. All of the curves look very similar! A reasonable baseline motivated by these results is to just apply a simple decay function to R(t) with a single hyperparameter controlling the rate of decay. I suspect this would also work better than the co-teaching approach, and perhaps render the more complex method here unnecessary. In fact, all of the gains associated with this method could just be due to co-teaching dropping far less examples as training progresses, as its decay rule isn't optimal.




**Experience Assessment:**

I do not know much about this area.

**Review Assessment: Checking Correctness Of Derivations And Theory:**

I assessed the sensibility of the derivations and theory.

**Review Assessment: Checking Correctness Of Experiments:**

I assessed the sensibility of the experiments.

**Review Assessment: Thoroughness In Paper Reading:**

I read the paper at least twice and used my best judgement in assessing the paper.

---

> ### Author Response · Authors · 2019-11-11
> **Reply to Reviewer#3 (part 1)**
>
> Thanks for your comments.
>
> Q1. Section 3.1 isn't very compelling to me. Experiments done on just CIFAR with two architectures and optimizers are certainly not sufficient to make any broad claims. I don't think this qualifies as a "contribution" of the paper.
>
> Please check the new Figure 1 in the revised PDF. We have enumerated more perspectives there, i.e.,
> 1) three datasets (i.e., CIFAR-10, CIFAR-100, MNIST) with three noisy types (i.e., symmetric 20%, symmetric 50%, pair-flip 45%)
> 2) three models in learning from noisy labels
> 3) three optimizers (i.e., SGD, RMSProp, Adam)
> 4) two more important hyperparameters for optimizers (i.e., batch size, learning rate)
> 5) STD for each learning curve (resulting from 5 different runs)
> These datasets, models, and optimizers are all popularly used in the noisy label literature (Jiang et al., 2018; Han et al., 2018; Chen et al., 2019; Yu et al., 2019).
>
> We have also clarified the first contribution to the revised version. The point is that: we want to show why R(T) is hard to design, as it correlates with the memorization effects, which is hard to quantize.
>
> Q2. Most practitioners use early stopping to halt training after the performance on the validation set drops.
>
> Thanks for the suggestion, "early stopping" is indeed a choice for practical usage.
> 1) Please see Q3, "held-out curve" is a better measurement than "early stopping" to evaluate the robustness of a method.
> 2) We also reported the performance with "early stopping" in Appendix B.1 of the revised version. As can be seen, the proposed method not only has a better "held-out curve," but also a better performance than "early stopping."
>
> Q3. Why should we care about the held-out curve after the maximum is reached?
>
> Thanks for the suggestion. This is a standard practice in the noisy label literature. We add an explanation in Section 4.2 of the revised version. The "held-out curve" is a better measurement than "early stop" to evaluate how a method is robust to noisy labels (Zhang et al., 2016; Arpit et al., 2017).
> 1) Ideally, if a method is robust to noisy labels, then its performance will increase with more training epochs (not to memorize noisy labels). Thus, if a method's held-out curve quick falls after reaching the maximum, then it means the method is NOT robust intrinsically.
> 2) If a method has a good "held-out curve," it is more likely to have better performance than "early stopping." This is also the case for the proposed approach.
> Finally, we also report results with the early stop in Appendix B.1 of the revised version.
>
> Q4. Shouldn't we care more about the training curve, as at some point during training, the noisy labels will also be memorized? Isn't this the definition of the "memorization effect"?
>
> No, memorization cannot be seen from the training loss.
> 1) Please see our introduction, and (Zhang et al., 2016; Arpit et al., 2017). Memorization means: "learn easy patterns first and then over-fit on (possibly noisy) training data set." This means the training loss with always gets smaller with more epochs, no matter there are noisy labels or not. Thus, we cannot see memorization from the training curve.
> 2) Memorization must be seen from the "held-out curve," which will increase first and then significantly decrease resulting from the memorization of noisy labels. This is also why the "held-out curve" is a good measurement (please see Q3).
> We have also shown what is the memorization effect in the revised version, i.e., top of page 2 (Section 1 Introduction) and Figure 1(a-b).
>
> Q5. What about standard baseline methods, e.g., active learning to help with this problem? Active learning seems highly relevant, yet it is not mentioned anywhere in this paper.
>
> Thanks for the suggestion. We have added such a discussion in the revised version in Section 2.1. Active learning is not applicable here (see Active Learning Literature Survey, Burr Settles):
> 1) To do active learning, we need to obtain a classifier of which the performance is good enough to generate confidence predictions.
> 2) Active learning is sensitive to noisy labels and outliers.
> Thus, active learning is a choice to get more labeled data when there are only a few high-quality ones, not applicable for directly learning from noisy labels here.

---

> > ### Author Response · Authors · 2019-11-11
> > **Reply to Reviewer#3 (part 2)**
> >
> > Q6. Are all of the basic functions in Fig 2 necessary for the performance of the proposed method? How were they selected?
> >
> > We do not select, all of them (in Figure 2) are used, and they are all necessary. A comparison is in Appendix B.2, which shows a simple decay function is not good enough (please also see Q9).
> >
> > Q7. Why is this motivated by the Taylor expansion?
> >
> > Thanks for the suggestion. We have updated our explanation in the revised version. We want to show R(t) can be approximated by a group of basis functions.
> >
> > Q8. All of the curves look very similar.
> >
> > Thanks for the comments. In the revised version, we have updated Figure 5. The searched R(T) enjoys much diversity now.
> >
> > Q9. A reasonable baseline motivated by these results is to apply a simple decay function to R(t) with a single hyperparameter controlling the rate of decay.
> >
> > Thanks for the comments. We add such experiments in Appendix B.2 of the revised version.
> > We can see that performance obtained from the proposed method is much better than that from a simple decay function. This again demonstrates the needs of approximating R(T) by a linear combination of some basic functions.
> >
> > Q10. All of the gains associated with this method could just be due to co-teaching dropping far fewer examples as training progresses, as its decay rule isn't optimal.
> >
> > Thanks for pointing this out. Yes, the decay rule in origin Co-teaching is not optimal, but it is hard to design such R(T):
> > 1) Searching R(T) is not an easy problem as it correlates with the memorization effect, which is hard to quantize (see Figure 1).
> > 2) Please see Figure 5 in the updated version. We have updated experiments now, and we can see that the behavior of R(T) diverse and simply dropping more (also see Q9) does not work. For example, in the last row of Figure 5, all curves first decrease and then increase.
> > 3) Please also check Appendix B.3, label precision is significantly increased by the searched R(T), which means the quality of samples used for training is greatly improved.

---

### Official Review · AnonReviewer2 · 2019-10-30
**Official Blind Review #2**

**Rating:** 3

**Review:**

This paper focuses on the topic of learning from noisy -- or as they call it "corrupted" -- labels. Specifically this focuses on an approach where data selection -- ideally of cleaner/less noisy examples --  can help the learn model overcome data noise, akin to the approaches this builds upon (i.e, the Co-Teaching and MentorNet approaches). The specific idea here is to take an AutoML style approach to the problem  in particular to determine how many examples are selected in each mini-batch. The proposed method is based upon natural gradient based updates to the hparams (which was really the only feasible way to tackle this problem given the complex dependence on the hparams and a good choice). The experimental results using synthetic noise corruption are indicative of improved performance compared to the baseline techniques.

Overall while I thought the paper made for a very interesting read and showed some great promise I had some significant concerns as well.

On the plus side:

+ The empirical results on the simulated noisy data are quite positive/
+ The proposed method makes sense as does the search algorithm in the hparam space.

My main concerns with the work stem from the empirical study and choices made there. While I understand that other existing techniques like the Co-teaching and MentorNet approaches have used simulated noise to study the impact of performance of these robustness techniques, at some point I question their validity on real datasets. Noise patterns in real datasets hardly follow some set pattern and thus I hesitate to read much into results derived solely on synthetic datasets. Given that the goal of these techniques is to improve performance when training with real, noisy labeled data why not actually demonstrate performance on such benchmarks? For example, there are numerous datasets from domains like crowdsourcing that allow you to get "noisy" ratings for datapoints. Wouldn't a more compelling argument be derived by showing improved performance on such datasets?

Thus to summarize: I worry that the results derived solely on simulated noise may not be very indicative of performance in more realistic settings and would request the authors to consider providing evidence on more realistic datasets.

I also wanted to note that the paper exposition is lacking in some aspects and I needed to reread certain sections to make sure I understood them correctly. I think the paper would benefit from a good proofread not just from the grammar/spelling perspective (which there are multiple instances which could be improved) but also from the overall presentation and legibility perspective.

All this said: I want to clarify that this topic is not my research focus and hence I am uncertain as to how much findings on these simulated noise patterns carry over to real datasets and their associated noise patterns. If there is existing evidence indicating strong correlation, then perhaps my review may have varied.

**Experience Assessment:**

I do not know much about this area.

**Review Assessment: Checking Correctness Of Derivations And Theory:**

I assessed the sensibility of the derivations and theory.

**Review Assessment: Checking Correctness Of Experiments:**

I assessed the sensibility of the experiments.

**Review Assessment: Thoroughness In Paper Reading:**

I read the paper at least twice and used my best judgement in assessing the paper.

---

> ### Author Response · Authors · 2019-11-11
> **Reply to Reviewer#2**
>
> Thanks for your comments.
>
> Q1. Why not demonstrate performance on real datasets?
>
> Thanks for the suggestion. We are also aware of this potential problem, and we have done this part after the submission. Please check Section 4.3 in the revised PDF.
>
> Following (Wang et al., 2019), which is the new state-of-the-art deep face recognition method, we have tested the proposed method on face data sets. We train with VggFace2-R (Cao et al., 2018) data set, which is a noisy data set collected from the Google image search.
>
> Table 1 shows that the proposed method can consistently achieve the best performance on such real data sets. Thus, our method is not only useful with synthetic noise but also works well on real applications.
>
> Q2. I am uncertain as to how much findings on these simulated noise patterns carry over to real datasets and their associated noise patterns. If there is existing evidence indicating a strong correlation, then perhaps my review may have varied.
>
> Thanks for the question. Yes, they are strongly correlated. We add the explanation of the synthetic noise in Section 4.2 of the revised version. Specifically,
> 1) The controlling variable can justify the effectiveness of the proposed method under specific conditions. In the context of learning with corrupted labels, the noise pattern is regarded as the key controlling variable. There are several common noise patterns (Patrini et al., 2017, Han et al., 2018), such as symmetric-flip and pair-flip.
>
> All these noise patterns correspond to real-world scenarios. For example, on the macro-level, class cat flipping to the class dog makes sense, while class dog flipping to class cat also makes sense. Such flipping yields a noise pattern called symmetric-flip (Patrini et al., 2017). On the micro-level, for dogs, class Norfolk terrier flipping to class Norwich terrier makes sense, while class Norfolk terrier flipping to class Australian terrier not. This flipping yields a noise pattern called pair-flip (Han et al., 2018a), which depicts the fine-grained classification case.
>
> 2) Since the noise pattern of real-world datasets can be the combination of simple noise patterns, we should first verify whether our proposed method works well on several common noise patterns before delving into complex real-world datasets. This is quite common in the area of learning with corrupted labels.
>
> 3) Please also see Q1 above, the performance of the proposed method is therefore consistent on both synthetic and real data sets (Section 4.3).

---

### Author Response · Authors · 2019-11-11
**Common Questions to All Reviewers**

We thank all reviewers' efforts in this paper, here we summarize three common questions.

Q1. About the presentation of this paper.

We have significantly revised the paper in the updated version, and please check the uploaded PDF. Changes are all highlighted in blue, briefly,
1) make the definition of the memorization effect in the introduction more clear and show this in Figure 1.
2) re-write the first contribution, make it clear why it is hard to design R(T).
3) discuss the connection with active learning in Section 2.1.
4) emphasize the main concepts in AutoML and remove unnecessary ones, and clarify connections of the proposed method with AutoML in Section 2.2.
5) re-write and re-drawn all figures to make legend and axis more clear
6) remove the explanation on Taylor expansion, and re-write paragraphs around equation (1)
7) add an explanation of why the held-out curve is used as a measurement in Section 4.2.
8) explanation the needs of synthetic noise in Section 4.2.
9) add experiments on real applications on face recognition in Section 4.3.

Q2. The importance of searching R(T).

First, see (Han et al., 2018b, Jiang et al.,2018, Yu et al., 2019), the performance of sample-selection methods heavily depends on R(T). In this revised version, we have emphasized this point above Algorithm 1.

Next, please see Figure 1 in the revised version, since the memorization effect and R(T) correlates with the effect, it is difficult to design R(T) by hand. Such difficulties motivate us to solve this problem by AutoML.

Besides, in the revised version, we have fully updated our experiments, see Figure 5. R(t) represents a great diversity. Simply dropping more samples does not work, and the proposed method can efficiently search a proper R(t) for each problem (Figure 6).

We also add results on label precision to further explain why the seared curve can be better in Appendix B.3. We can see the searched R(T) can significantly improve the number of clean labels used for training.

Finally, in this way, Co-teaching with searched R(T) can even beat methods that use better criterions to find clean samples, i.e., Proposed v.s. Co-teaching+ (Yu et al., 2019) in Figure 4 on benchmark data sets and Proposed in Co-Mining (Wang et al., 2019) in Table 1 on the real data set.

Q3. Technical contributions.

The first technical contribution is
1) Show why R(T) is difficult to design (Figure 1): R(T) correlates with the memorization effect, which depends on many factors and hard to quantize (also see Q2). We also clarify this point in the first contribution of the revised version.

Based on the above observations, we
2) design a domain-specific search space to exploit the memorization effect.
3) propose a new method for hyperparameter optimization based on the analysis of problems on existing first order and derivative-free algorithms (Section 3.3.1).

For a summary of 2) and 3), please also see Reviewer 2 comments: "The proposed method is based upon natural gradient-based updates to the hparams (which was really the only feasible way to tackle this problem given the complex dependence on the hparams and a good choice)".

Besides, we also make the 2) and 3) clearer in this revised version. Please see the difference with existing AutoML techniques at the end of Section 2.2.

---

### Decision · Program_Chairs · 2019-12-19

**Decision:**

Reject

**Comment:**

This paper develops a method for sample selection that exploits the memorization effect. While the paper has been substantially improved from its original form, the paper still does not meet the quality bar of ICLR in terms of presentation of the results and experimental validation. The paper will benefit from a revision and resubmission to another venue.